UPDATE ARTICLE

# The *Pseudomonas aeruginosa* ribonuclease Ribocin cleaves eukaryotic ribosomes at helix 69 to inhibit host translation

Alejandro Vasquez-Rifo[1]*, Denis Susorov[2], Emily H. Sholi[2], Gabriel Demo[3], Yasaman Jami[4], Jihui Sha[4], James A. Wohlschlegel[4], Andrei Korostelev[2]*, Victor Ambros[1]*

**1** Program in Molecular Medicine, UMass Chan Medical School, Worcester, Massachusetts, United States of America, **2** RNA Therapeutics Institute, UMass Chan Medical School, Worcester, Massachusetts, United States of America, **3** Central European Institute of Technology, Masaryk University, Brno, Czech Republic, **4** Department of Biological Chemistry, David Geffen School of Medicine, University of California, Los Angeles, California, United States of America

\* alejandro.vasquezrifo@umassmed.edu (AV-R); andrei.korostelev@umassmed.edu (AK); victor.ambros@umassmed.edu (VA)

The Editors encourage authors to publish research updates to this article type. Please follow the link in the citation below to view any related articles.

## Abstract

*Pseudomonas aeruginosa* employs host translation inhibition as a virulence-enhancing strategy. We previously showed that the bacterium induces cleavage of *Caenorhabditis elegans* large ribosomal RNA at helix 69 (H69), part of a central intersubunit bridge and the ribosomal decoding center. In this study, we demonstrate that a previously uncharacterized ribonuclease, Ribocin, is necessary and sufficient for H69 cleavage. Recombinant Ribocin cuts H69 in worm and mammalian ribosomes, indicating that H69 cleavage by *P. aeruginosa* is phylogenetically conserved. In worms, mammalian cells, and rabbit reticulocyte lysates, H69 cleavage results in translation inhibition. Furthermore, Ribocin contributes to bacterial virulence toward *C. elegans*, triggers a major host response to translation inhibition, and operates in parallel with Exotoxin A-mediated translation inhibition. These findings unveil the first known nuclease that cleaves eukaryotic ribosomes at H69 and expand the understanding of host translation-inhibition by establishing targeted rRNA cleavage as a mechanism of host attack.

## Introduction

Protein synthesis—an essential process for gene expression and cell growth—is a primary target of antagonistic ecological interactions. For example, the fungal sarcin and plant ricin toxins inactivate ribosomes by cleaving (sarcin) or depurinating (ricin) the essential and conserved sarcin—ricin loop (H95) of large ribosomal RNA, acting as anti-predator strategies [1,2]. Pathogenic bacteria can also depurinate H95 (e.g., Shiga toxin [3]) or inhibit eukaryotic translation by inactivating host translation factors [4] (e.g., Diphtheria toxin, Exotoxin A). Lastly, bacterial protein synthesis is targeted

**Data availability statement:** All data is available in the manuscript and Supporting information files. Raw image data can be found in S1 Raw Images.

**Funding:** This research was supported by funding from the US National Institutes of Health (https://www.nih.gov) R35GM127094 to A.A.K., R35GM131741 to V.A. and R35 GM153408 to J.A.W. and F31HL180041 to E.H.S., and from the Pew Charitable Trusts (https://www.pew.org) 00027360 to A.V-R. The funders did not play any role in the study design, data collection and analysis, decision to publish, or preparation of the manuscript.

**Competing interests:** The authors have declared that no competing interests exist.

**Abbreviations:** AS, ammonium sulfate; CDI, contact-dependent growth inhibition; cDNA, complementary DNA; CMMB, carboxylate-modified magnetic beads; CV, column volumes; FBS, fetal bovine serum; H69, helix 69; IC$_{50}$, half-maximal inhibitory concentration; LB, Lysogeny Broth; PDB, Protein Data Bank; qPCR, real-time quantitative PCR; QS, quorum sensing; RRL, rabbit reticulocyte lysates; SK, slow killing; TFP, tandem fractionation procedure.

in microbial interactions. For example, bacterial colicin E3 and CdiA-CT$^{ECL}$ nucleases attack other bacteria's ribosomes by cleaving small subunit rRNA at h44 [5–7], and alternative strategies rely on small molecules or peptides that inhibit translation by binding to the ribosome or translation factors [8,9]. Discovering and understanding the strategies of microbe-induced inhibition of host translation is critical for elucidating the mechanisms of disease pathogenesis and developing antimicrobial therapies.

*Pseudomonas aeruginosa* is a bacterium that infects a wide range of hosts. In humans, it causes potentially fatal infections, such as hospital-acquired pneumonia, which are particularly severe in immunocompromised patients [10,11]. To investigate these infections, the interaction between *P. aeruginosa* PA14 and the nematode *C. elegans* serves as an established model system that has provided insights into host immunity and bacterial pathogenesis. Under "slow killing" (SK) co-culture conditions, bacteria colonize the intestines of adult worms and kill them over the course of approximately 3 days [12]. Under these conditions, the bacteria express virulence-promoting gene expression programs through quorum-sensing (QS) regulatory pathways [13,14]. In response, the worms activate multiple behavioral, stress-response, and innate immune pathways [15–17].

*P. aeruginosa* potently induces host translation inhibition. A well-studied strategy is via the Exotoxin A protein (ToxA), which ADP-ribosylates eukaryotic elongation factor 2 [18–20]. However, ToxA exerts minimal effects during infection of *Caenorhabditis elegans* [21,22]. We recently reported evidence suggesting a second translation inhibition strategy [23]. Specifically, upon interaction with virulent *P. aeruginosa*, *C. elegans* experiences a rapid accumulation of ribosomes with 26S large ribosomal RNA cleaved at the loop of helix 69 (H69). H69 forms part of the ribosomal decoding center and is critical for translation [24–26]. In previous studies [23,27], we showed that H69 cleavage is regulated by QS, promoted by the R-body gene cluster, and opposed by host-response pathways. However, the H69 nuclease remained unknown, and it was unclear whether the nuclease was encoded by the host or the bacterium [23]. In this study, we identify and mechanistically characterize the H69 nuclease. We find that a *P. aeruginosa* nuclease, which we named Ribocin, cleaves eukaryotic ribosomes to inhibit host translation, activate a host-response to translation inhibition, and promote virulence in a bacterium–animal pathogenic interaction.

## Results

### Identification of H69 nuclease candidates

S100 lysates from infected worms—but not from uninfected worms—contain a factor that cleaves H69 in *C. elegans* ribosomes [23]. To identify the H69 nuclease, we established an in vitro H69-cleavage assay and performed two different tandem fractionation procedures, each yielding a single peak fraction with H69-cleavage activity (Fig 1A). We then used label-free LC–MS/MS to measure protein composition and abundance in fractions with peak H69-cleavage activity and in several adjacent fractions with intermediate or no cleavage activity. This analysis identified 230 candidate proteins enriched by both fractionation procedures (Fig 1A). A score-based metric quantified the correlation between candidate protein abundance and H69-cleavage

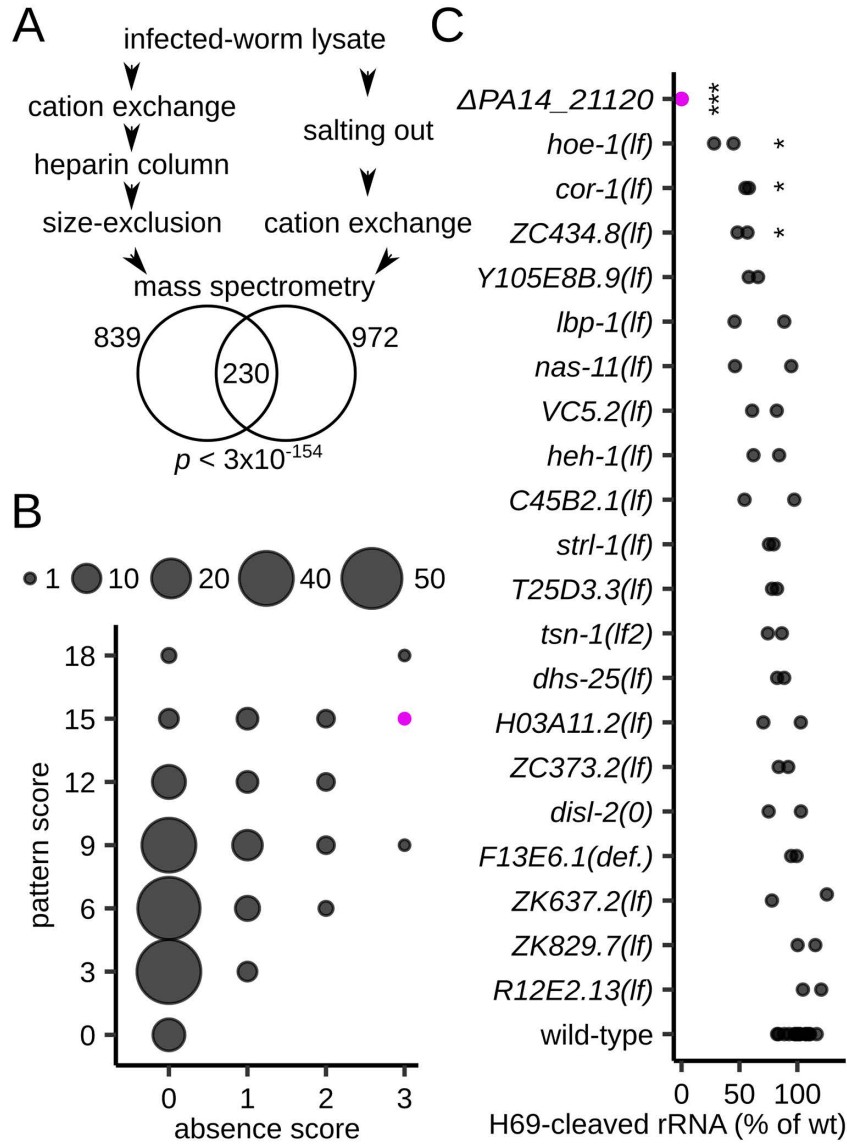

**Fig 1. Identification of H69 nuclease candidates. (A)** Scheme of the two tandem fractionation procedures. Venn diagram showing the number of protein candidates identified in each procedure and their overlap; *p*-value from the hypergeometric test is shown. **(B)** Scatterplot of absence and pattern scores for the 230 candidate nuclease proteins in **(A)**. Circle size denotes the number of candidates at each score combination. Higher scores indicate a better fit to expectation. The PA14_21120 candidate is shown in purple. **(C)** Boxplot of H69 cleavage levels for mutants of 21 candidate proteins; $n \geq 2$ biological replicates. The data underlying this figure can be found in S1 Data. "*" indicates a *p*-value < 0.05 and "***" indicates a *p*-value < 0.001, for two-sided Welch *t* test comparisons between mutant and wild-type (wt) control. All mutants are for *Caenorhabditis elegans* genes except for *PA14_21120*.

activity in peak and adjacent fractions and was used to prioritize proteins for further testing (Fig 1B and S1 Table; see Materials and methods).

For 21 high-ranking candidates, we identified publicly available *C. elegans* or PA14 mutants and analyzed their effects on H69 cleavage in *C. elegans* infected with *P. aeruginosa* PA14 (Fig 1C). Compared to wild-type worms, loss of function of three worm genes—encoding the coronin-like protein COR-1, the endonuclease HOE-1, and the putative kinase ZC434.8—reduced H69 cleavage by ~40%–60% (*p* < 0.05; Fig 1C) via unknown mechanisms. One PA14 mutant,

*ΔPA14_21120*—a null allele of an uncharacterized gene—reduced H69 cleavage by 100% (Fig 1C). Indeed, whereas H69 cleavage was observed within 24 h in worms infected with wild-type PA14, no H69 cleavage was detected over a 3-day infection time course with the *ΔPA14_21120* mutant (Fig 2A and 2B). These results demonstrate that the PA14_21120 protein is essential for the cleavage of *C. elegans* 26S rRNA at H69.

## RbcN is a predicted nuclease that is required for ribosome cleavage at H69

The *PA14_21120* gene is conserved among *P. aeruginosa* strains, possesses uncharacterized homologs primarily within the phylum Pseudomonadota, and encodes a protein with no annotated domains (S1A, S1B, and S2 Figs and S2 Table). However, the AlphaFold database [28] predicts a high-confidence PA14_21120 protein structure (average pLDDT 87.62; S1C Fig). Using the structure comparison Dali server [29], we determined that the closest structural homologs to PA14_21120 are predominantly RNase A superfamily nucleases (S3 Table). A structural alignment of PA14_21120 with its closest structure—the bacterial CdiA-CT^YKris nuclease [30]—shows high structural conservation despite low sequence

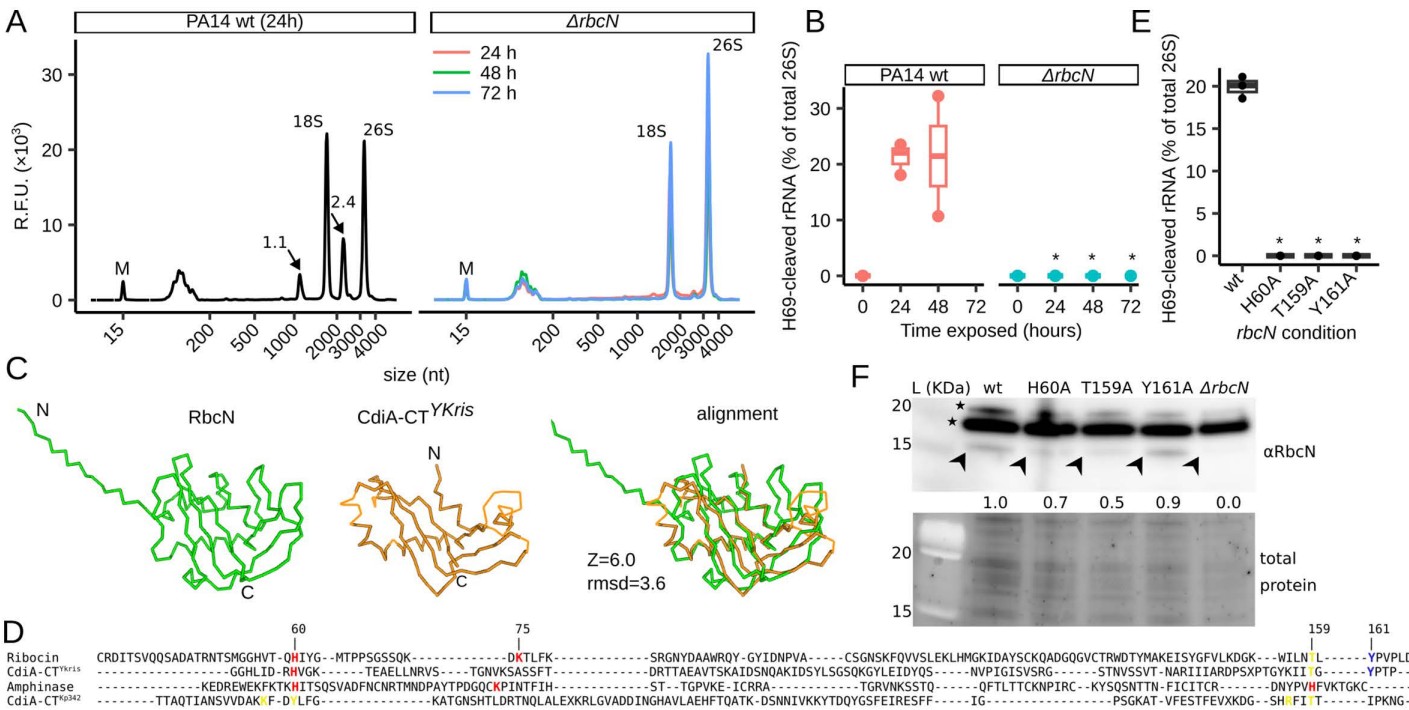

**Fig 2. Effect of *rbcN* mutants on ribosome cleavage and features of the RbcN protein. (A)** Total RNA profiles of live worms exposed to PA14 wild-type (wt) for 24 h or PA14 Δ*rbcN* for 24, 48, and 72 h. H69-cleaved rRNA fragments are indicated by arrows with their estimated nucleotide size (×10³). "M" indicates a 15-nucleotide (nt) marker. R.F.U. represents relative fluorescence units. **(B)** Boxplot of H69 cleavage levels for worms exposed to PA14 wt or PA14 Δ*rbcN* for 24 to 72 hours; *n*=2 biological replicates). **(C)** Comparison of an Alphafold model of RbcN (green) and the bacterial RNase CdiA-CT^YKris crystal structure (orange; PDB: 5E3E). The proteins' C- and N-termini are labeled. "Z" denotes Z-score. "rmsd" is root-mean-square deviation of the aligned structures. **(D)** Structure-guided protein sequence alignment of RbcN and three nucleases (CdiA-CT^YKris, Amphinase, and CdiA-CT^Kp342). Conserved amino acids experimentally shown to be required for RNA cleavage are color-coded as follows: red for structurally aligned in RbcN and Amphinase, yellow for RbcN and CdiA-CT^Kp342, and blue for RbcN and CdiA-CT^YKris. Residue numbers are shown for RbcN. **(E)** Boxplot of H69 cleavage levels for worms exposed for 24 h to PA14 wild-type (wt) or three *rbcN* point mutants (H60A, T159A, and Y161A); *n*=3 biological replicates. **(F)** western blot for RbcN abundance in PA14 wt, Δ*rbcN*, and the three shown *rbcN* point mutants. The arrowhead indicates RbcN, and the stars point to non-specific bands. RbcN abundance—normalized to total protein signal—is indicated by the numbers below each western blot lane. The data underlying this figure can be found in S1 Data. "*" indicates a *p*-value < 0.05 for two-sided Welch *t* test comparison to the wild-type control. "L" denotes the protein ladder.

homology (Fig 2C). We therefore tentatively renamed the *PA14_21120* gene as ribosome-cleaving and inactivating nuclease *(rbcN)* and its protein as Ribocin (RbcN).

To test whether Ribocin has intrinsic ribonuclease activity, we identified conserved amino acid residues positioned similarly to catalytic residues of RNase A superfamily nucleases that are critical for activity [30–33]. Structure-guided alignment of RbcN to three nucleases with the highest structural similarity (CdiA-CT$^{YKris}$, Amphinase, and CdiA-CT$^{Kp342}$) suggested that H60, K75, T159, and Y161 might contribute to the catalytic activity of RbcN (Figs 2D and S1D). To test this, we generated three *rbcN* mutants in which alanine substituted for H60, T159, or Y161. In worms exposed to PA14 encoding *rbcN* point mutants H60A, T159A, or Y161A, we detected no H69 cleavage (Fig 2E). Western blot analysis revealed that the levels of H60A, T159A, and Y161A protein were ~50%–90% of wild-type RbcN (Fig 2F), indicating that protein level does not account for the complete loss of H69 cleavage. Collectively, these results suggest that the PA14 *rbcN* gene encodes a ribonuclease necessary for ribosome cleavage at H69 in *C. elegans*.

## RbcN is sufficient to induce ribosome cleavage at H69 in worms and mammals

To determine whether RbcN is sufficient to induce H69 cleavage of *C. elegans* ribosomes in vivo, we transformed *E. coli*—which neither encodes an RbcN ortholog nor induces cleavage of worm 26S rRNA—with plasmids to express *P. aeruginosa* RbcN wild-type or H60A from an inducible promoter and fed the induced bacterial strains to worms. Notably, expression of *P. aeruginosa* Ribocin in *E. coli* induced robust H69 ribosome cleavage (Fig 3A). Cleavage of 26S rRNA depended entirely on the expression of Ribocin and the putative catalytic H60 residue (Fig 3A). These results indicate that Ribocin-induced H69 cleavage in *C. elegans* does not require additional factors specific to *P. aeruginosa*.

To further test whether RbcN is sufficient for H69 cleavage, we purified recombinant wild-type or H60A-mutant RbcN (S3A and S3B Fig) and tested whether they cleave 28S rRNA in rabbit reticulocyte lysates (RRL), a widely used system to study translation. As in worms, the large 28S rRNA was cleaved at H69—determined by band excision, cloning, and sequencing (Fig 3B and 3C)—in the presence of wild-type Ribocin, and cleavage depended entirely on the putative catalytic H60 residue (Fig 3B and 3C). Thus, Ribocin alone induces cleavage of mammalian ribosomes at H69 in RRL, without requiring additional bacterial factors.

Lastly, to determine whether Ribocin induces cleavage of human ribosomes inside cells, we electroporated 293T cells with recombinant wild-type or H60A Ribocin protein and analyzed rRNA profiles three hours after RbcN delivery. Wild-type Ribocin induced cleavage of human 28S rRNA in cells (Fig 3D). These results show that Ribocin-dependent cleavage of large rRNA in living eukaryotic cells is conserved between *C. elegans* and mammals. Altogether, Ribocin is sufficient for H69 cleavage in four distinct experimental contexts: *C. elegans* fed with *P. aeruginosa* PA14 or RbcN-expressing *E. coli*, and rabbit reticulocyte lysates or mammalian cells treated with recombinant Ribocin.

The nuclease activity of RbcN appears to be directed specifically toward eukaryotic H69 in large rRNA. Purified recombinant RbcN did not detectably cleave an in vitro-transcribed mRNA (S3C Fig), and RNA profiles from RbcN-treated RRL ribosomes or human cells did not reveal major degradation products besides the two 26S rRNA fragments expected from cleavage at H69 (Fig 3B and 3D). Moreover, recombinant Ribocin did not cleave *E. coli* 70S ribosomes in translation-competent S30 extracts (S4 Fig). These results support the conclusion that RbcN specifically cleaves eukaryotic H69 and exhibits minimal non-specific nuclease activity.

## Ribosome cleavage at H69 impairs translation

To examine the effects of H69 cleavage on translation, we measured how the addition of recombinant Ribocin affects translation of a nanoluciferase reporter mRNA in RRL [34]. Preincubation of RRL with wild-type Ribocin resulted in a dose-dependent inhibition of nanoluciferase activity, with a half-maximal inhibitory concentration (IC$_{50}$) of 33 nM (Fig 4A and 4B), similar to those of angiogenin and other RNases that potently repress translation [35–37]. By contrast,

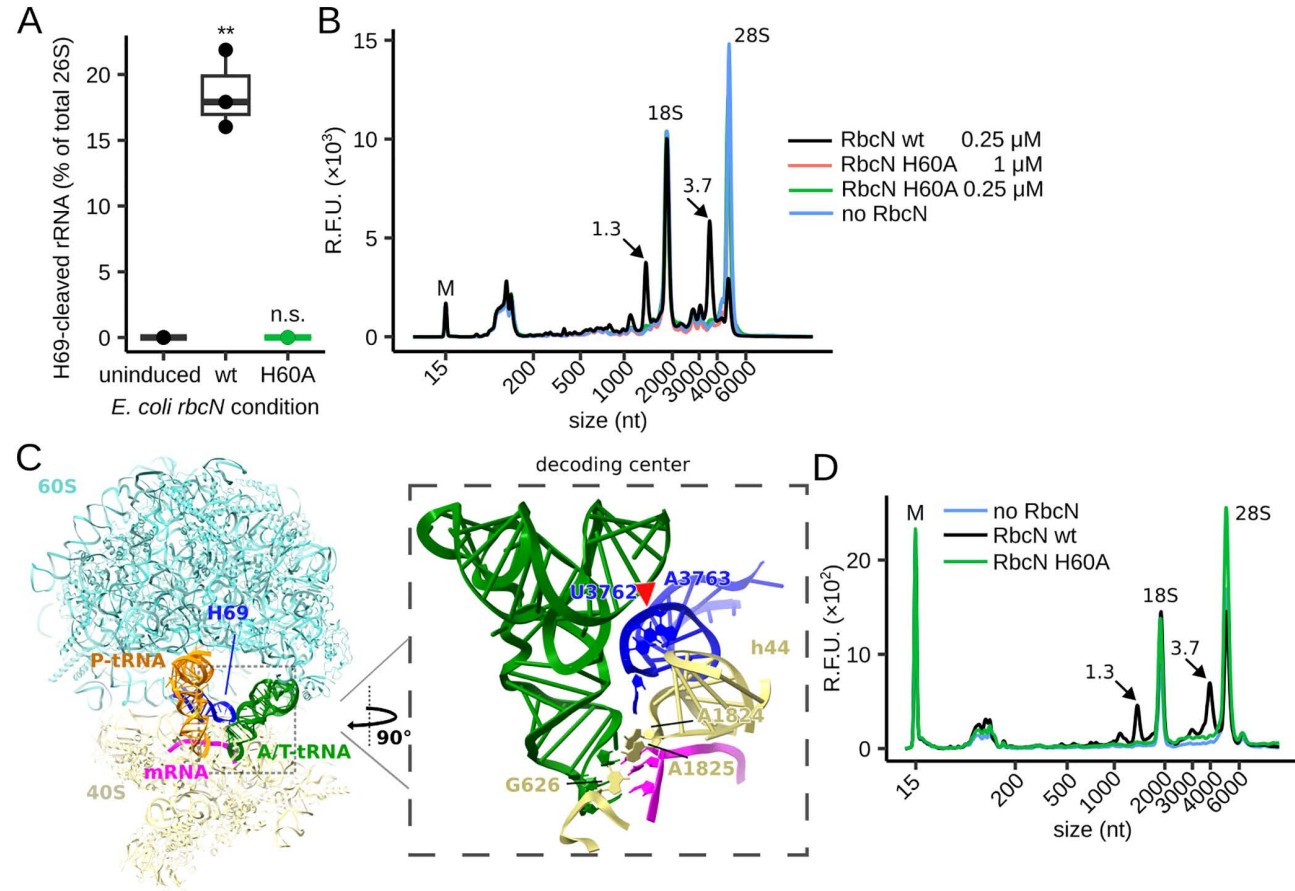

**Fig 3. Effects of *E. coli* expressing *rbcN* on *Caenorhabditis elegans* ribosomes and recombinant RbcN on mammalian ribosomes. (A)** Boxplot of H69 cleavage levels for worms exposed to *E. coli* expressing the *rbcN* wild-type (wt) or the *rbcN*(H60A) mutant gene for 12 hours (h). Uninduced *E. coli* with the *rbcN* wt construct was used as the control condition (uninduced); $n = 3$ biological replicates. **(B)** RNA profiles of ribosomes in rabbit reticulocyte lysates (RRL) after treatment with recombinant wild-type RbcN (wt) or H60A mutant RbcN at the indicated concentrations. H69-cleaved rRNA fragments are indicated by arrows with their estimated nucleotide sizes (×10³). The colored lines largely superimpose. **(C)** right and left: Structure of the decoding center in the rabbit 80S ribosome (PDB: 5LZS). The location of H69 cleavage by RbcN, determined from the wt RbcN sample in (B), is indicated by the red triangle pointing between U3762 and A3763 (rabbit 28S rRNA numbering). **(D)** Overlap of total RNA profiles of human 293T cells 3 h post-electroporation with wild-type RbcN (wt) or H60A mutant. H69-cleaved rRNA fragments are indicated by arrows with their estimated nucleotide size (×10³). The data underlying this figure can be found in S1 Data. "M" indicates a 15-nucleotide (nt) marker. R.F.U. represents relative fluorescence units. "**" indicates a p-value < 0.01 for two-sided Welch t test comparison to the uninduced condition (A). "n.s." denotes not significantly different.

preincubation with H60A-mutant Ribocin did not significantly inhibit translation, indicating that the strong translation-inhibitory activity of Ribocin depends on H69 cleavage.

To test whether Ribocin-mediated H69 cleavage of 28S rRNA inhibits translation in human cell culture, we co-electroporated 293T cells with nanoluciferase mRNA in the presence or absence of Ribocin. Indeed, wild-type Ribocin—but not the H60A mutant—suppressed nanoluciferase activity (Fig 4C), indicating that H69 cleavage can inhibit translation in human cells.

To evaluate the effects of RbcN on translation in *C. elegans*, we built a transgenic strain (*vha-6p::mScarlet-I3::pest::tbb-2* 3'UTR) in which the fluorescent mScarlet-I3 protein is tagged with a PEST degron and expressed in the intestine, the main organ to experience H69 cleavage [23]. Given the rapid maturation and degradation of mScarlet-I3 protein [38–40],

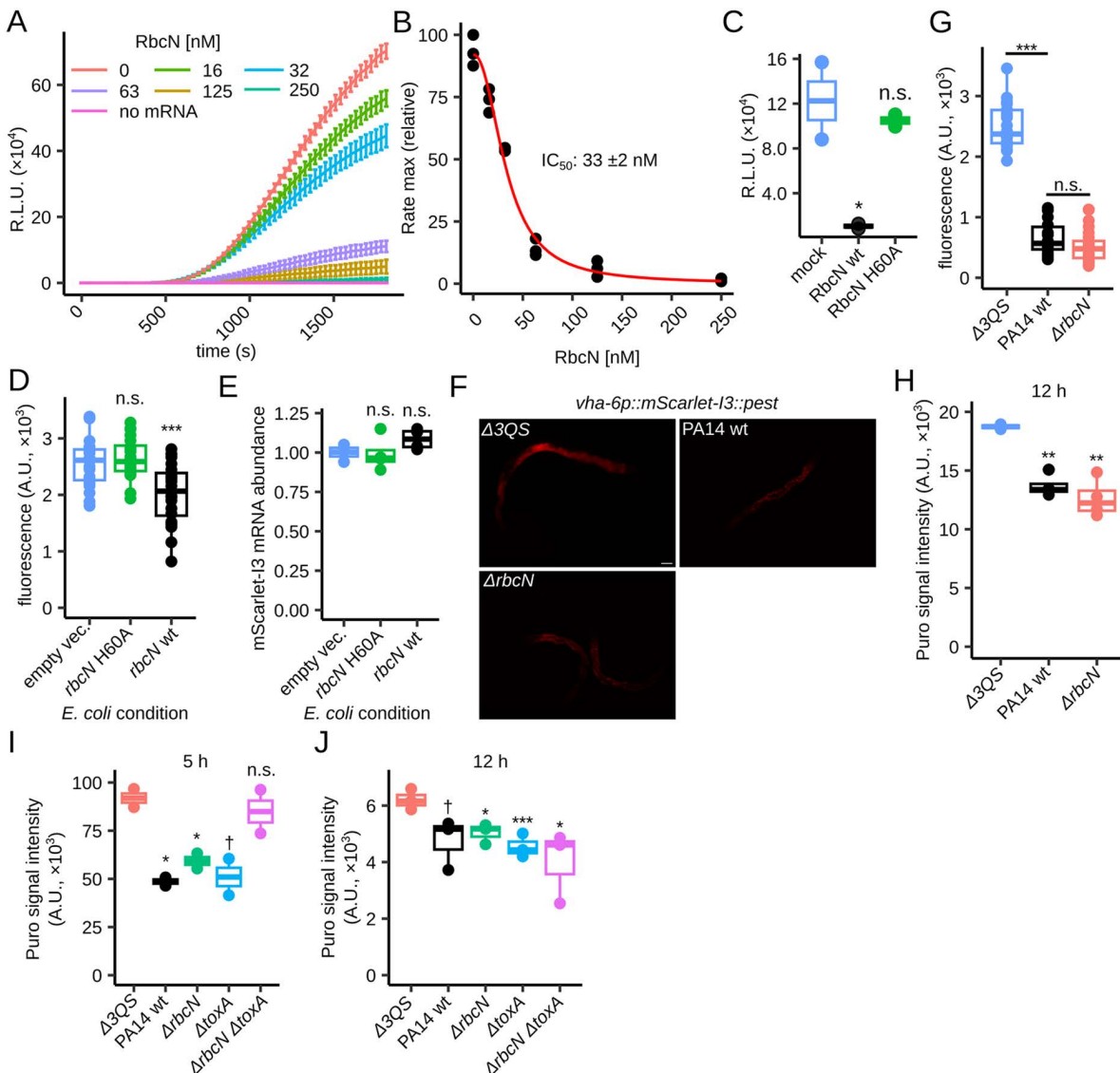

**Fig 4. Effects of RbcN/*rbcN* on translation. (A)** Luciferase-mediated in vitro translation assays. Luminescence of translated nanoluciferase mRNA upon exposure to increasing concentrations of wild-type RbcN protein (0–250 nM). Data are presented as mean ± SEM; $n \geq 2$ independent replicates. **(B)** Dose-response curve for the in vitro translation experiment shown in (A). The half-maximal inhibitory concentration (IC$_{50}$) is indicated. **(C)** Boxplot of relative luminescence in human 293T cells co-electroporated with nanoluciferase mRNA in the presence of recombinant RbcN—wild-type (wt) or H60A—or in the absence of RbcN (mock); $n = 2$ biological replicates. **(D)** Boxplot of the mean mScarlet-I3::PEST mean fluorescence intensity for worms exposed to *E. coli* heterologously expressing: *rbcN* H60A, *rbcN* wt, or "empty-vector" control; $n = 20$ individual animals. **(E)** Boxplot of the relative abundance of the *mScarlet-I3::pest* mRNA measured by RT-qPCR for worms in the conditions shown in (D); $n = 4$ biological replicates. **(F)** Fluorescent microscopy images of *mScarlet-I3::pest* worms (strain VT4414) exposed to PA14 strains for 12 hours. Scale bar measures 50 µm for all images. **(G)**. Boxplot of the mean mScarlet-I3::PEST fluorescence intensity for worms in the conditions shown in (F). $n = 20$–23 individual animals. **(H)** Boxplot of quantified anti-puromycin western blot for worms exposed to the wild-type PA14 strain (wt), Δ*rbcN*, or avirulent Δ*rhlR* Δ*lasR* Δ*mvfR* (Δ3QS) for 12 hours. The tubulin-normalized mean signal intensity for the anti-puromycin antibody (Puro) is indicated; $n = 2$ or 4 biological replicates. **(I, J)** Boxplot of quantified anti-puromycin western blot for worms exposed to the wild-type PA14 strain (wt), or Δ*rbcN*, or Δ*toxA*, or avirulent Δ*rhlR* Δ*lasR* Δ*mvfR* (Δ3QS) for 5 hours (I) or 12 hours (J). The tubulin-normalized mean signal intensity for the anti-puromycin antibody (Puro) is indicated; $n = 2$ biological replicates in (I); $n = 3$ biological replicates in (J). The data underlying this figure can be found in S1 Data. R.L.U. represents relative luminescence units. A.U. denotes arbitrary units. "n.s." denotes not significantly different. "**" indicates a $p$-value < 0.01 and "***" indicates a $p$-value < 0.001 for two-sided Welch $t$ test comparisons.

we reasoned that the mScarlet-I3 reporter would enable a proxy measurement of translation. Indeed, incubating worms in cycloheximide for 3.5 hours reduced mScarlet-I3 fluorescence without significantly reducing *mScarlet-I3* mRNA level (S5A and S5B Fig). We then exposed mScarlet-I3 reporter worms to *E. coli* expressing wild-type or H60A RbcN for various times before measuring fluorescence intensity. A 3-hour exposure to *E. coli* expressing wild-type RbcN—but not the H60A mutant—reduced mScarlet-I3 fluorescence in worms by ~25%, without reducing *mScarlet-I3* mRNA abundance (Figs 4D, 4E, and S5C). These results indicate that H69 cleavage inhibits translation in worms, after a short-term exposure to RbcN. At 7 hours, however, although wild-type RbcN significantly reduced mScarlet-I3 fluorescence compared to H60A RbcN, the reduction in fluorescence coincided with a reduction in *mScarlet-I3* mRNA (S5D and S5E Fig). We detected no change in pre-mRNA levels (S5E Fig), suggesting that the reduced mRNA level is caused by mRNA destabilization, perhaps as a stress response induced by translation inhibition. Consistent with this possibility, of four other intestine-specific genes that we analyzed, three showed reduced mRNA abundance after 7 h exposure to RbcN (S5F Fig).

To determine whether Ribocin is required for translation inhibition during infection of *C. elegans* by *P. aeruginosa*, we exposed mScarlet-I3 reporter worms to wild-type PA14, Δ*rbcN* mutant, or an avirulent mutant Δ*rhlR* Δ*lasR* Δ*mvfR (Δ3QS)* [14]. Compared to the avirulent control strain, both wild-type and Δ*rbcN* PA14 caused similar strong reductions in mScarlet-I3 fluorescence after 12-hour exposure (Fig 4F and 4G), when H69 cleavage is plentiful for PA14-exposed worms [23]. We next tested the effect of Ribocin on global translation. Using a puromycin incorporation assay, we found—consistent with the reporter-based findings—that after 12 h wild-type PA14 and Δ*rbcN* caused similar levels of translation inhibition in worms (Fig 4H). These results indicate that PA14 can trigger translation inhibition independently of Ribocin, suggesting the involvement of additional translational inhibitors.

*P. aeruginosa* also inhibits host translation by deploying the ToxA protein [18–20]. To determine the relative contributions of RbcN and ToxA to translation inhibition, we generated a Δ*toxA* single mutant and a Δ*toxA* Δ*rbcN* double mutant. Using the puromycin assay, we observed that after a 5-hour exposure, PA14 wild-type inhibited translation in *C. elegans* by ~50% compared to the avirulent Δ*3QS* strain (Fig 4I). The Δ*rbcN* and Δ*toxA* single mutants also inhibited translation by ~50%, but the Δ*rbcN* Δ*toxA* double mutant did not elicit any inhibition (Fig 4I). These results suggest that Ribocin and ToxA provide redundant contributions to PA14-mediated inhibition of host translation during the first 5 hours of exposure to worms. By 12 hours of infection, however, the wild-type PA14, Δ*rbcN,* and Δ*rbcN* Δ*toxA* strains caused similar levels of translation inhibition in worms (Fig 4J). Collectively, RRL assays, cell electroporation and the worm reporter and infection setups demonstrate that RbcN-mediated H69 cleavage induces translation inhibition. Upon infection, RbcN and ToxA induce translation inhibition at early stages of infection, whereas additional mechanisms, such as transcriptional regulation, may contribute to *P. aeruginosa*-induced gene expression repression at later stages.

## Ribocin contributes to full bacterial virulence and induces a host response to translation inhibition

The above findings suggest that Ribocin inhibits host protein synthesis as part of a bacterial strategy to oppose host defense mechanisms. To assess the contribution of *rbcN* during PA14 infection, we compared worm survival to wild-type PA14 versus Δ*rbcN*. Loss of *rbcN* extended the survival of worms exposed to PA14 by a small but significant margin (Figs 5A and S6A Fig), indicating that RbcN promotes full bacterial virulence in *C. elegans*.

The ZIP-2/IRG-1 pathway is a major infection response pathway in *C. elegans* stimulated by translation inhibition [21,22,41]. We found that an *irg-1::GFP* reporter transgene in worms was strongly activated after 10-hour exposure to wild-type PA14. However, expression of *irg-1::GFP* was ~95% lower in worms exposed to Δ*rbcN* (Fig 5B and 5C). Moreover, RbcN was sufficient to activate the ZIP-2/IRG-1 pathway, as the *irg-1::GFP* reporter transgene was activated by *E. coli* expressing wild-type RbcN but not the H60A mutant (Figs 5D and S7). These results indicate that RbcN activity contributes measurably to PA14 virulence and that RbcN-mediated translation inhibition is largely responsible for ZIP-2/IRG-1 pathway activation in PA14-infected worms.

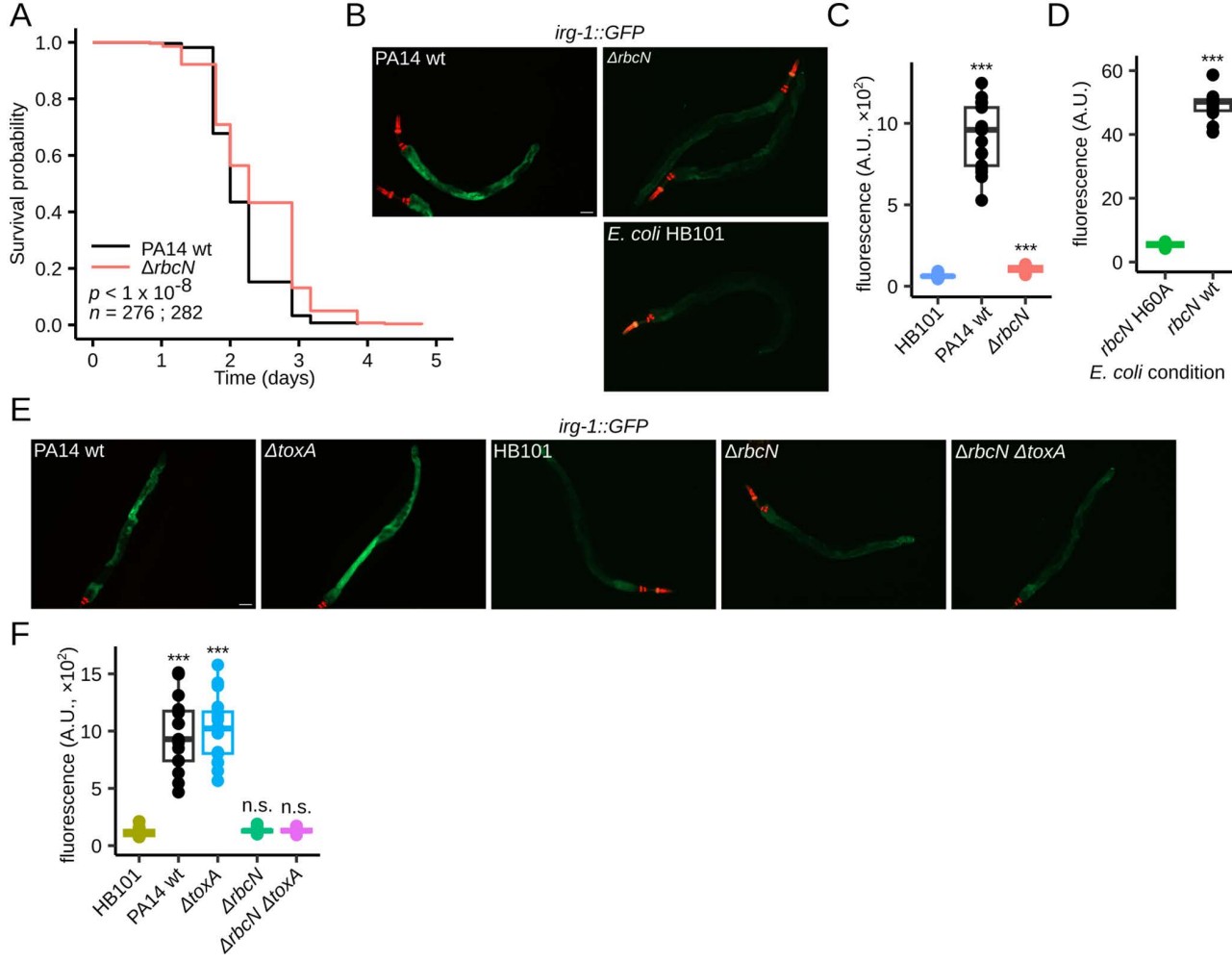

**Fig 5. Effect of *rbcN* and *toxA* on virulence and the ZIP-2/IRG-1 pathway. (A)** Survival curve of adult worms exposed to PA14 wild-type (wt, in red) or Δ*rbcN* (in blue). The *p*-values from Tarone–Ware and log-rank tests and sample sizes (*n*, individual animals) are shown. **(B)** Fluorescence microscopy images of worms expressing *irg-1::GFP* exposed to *E. coli* HB101 (control condition), PA14 wild-type (wt), and PA14 Δ*rbcN*. Scale bar measures 50 μm for all images in (B, E). **(C)** Boxplot of the mean intestinal fluorescence (GFP channel) for worms in the conditions shown in (B): HB101 (*n* = 9), PA14 wt (*n* = 17), Δ*rbcN* (*n* = 18) individual animals. **(D)** Boxplot of the mean intestinal fluorescence (GFP channel) for worms exposed to *E. coli* expressing: *rbcN* H60A (*n* = 8), or *rbcN* wt (*n* = 11) individual animals. **(E)** Fluorescent microscopy images of *irg-1::GFP* worms exposed to *E. coli* HB101 (control condition), PA14 wt, Δ*rbcN*, Δ*toxA*, Δ*rbcN* Δ*toxA* for 12 hours. **(F)** Boxplot of the mean intestinal fluorescence (GFP channel) for worms in the conditions shown in (E): HB101(*n* = 20), PA14 wt (*n* = 15), Δ*rbcN* (*n* = 20), Δ*toxA* (*n* = 17), Δ*rbcN* Δ*toxA* (*n* = 21) individual animals. The data underlying this figure can be found in S1 Data. A.U. denotes arbitrary units. "n.s." indicates not significant. "†" indicates *p* = 0.06, "*" represents a *p*-value < 0.05, "**" represents a *p*-value < 0.01 and "***" represents a *p*-value < 0.001 for two-sided Welch *t* test comparisons.

## Ribocin mediates stronger effects than ToxA on virulence and the ZIP-2/IRG-1 pathway

To examine the relative contributions of RbcN and ToxA to PA14 virulence and ZIP-2/IRG-1 pathway induction, we determined worm survival and *irg-1::GFP* induction in Δ*rbcN*, Δ*toxA,* and Δ*rbcN* Δ*toxA* mutants. As previously reported [22], worm survival was identical between the Δ*toxA* mutant and wild-type PA14 bacteria (S6B Fig), and the Δ*toxA* Δ*rbcN* double mutant was as virulent to worms as the Δ*rbcN* mutant (S6C and S6D Fig). Moreover, Δ*toxA* did not affect *irg-1::GFP* activation in the presence or absence of RbcN (Fig 5E and 5F). These results suggest that, unlike RbcN, ToxA does not contribute measurably to PA14 virulence or to activation of the ZIP-2/IRG-1 pathway.

## Discussion

In this study, we identified Ribocin as the bacterial nuclease responsible for cleaving host's ribosomes at H69 during the infection of *C. elegans* by *P. aeruginosa*. Ribocin is a previously uncharacterized nuclease, whose structural fold is homologous to RNase A superfamily proteins from bacteria (CdiA-CT[YKris] and CdiA-CT[Kp342]) and vertebrates (Amphinase). However, Ribocin's sequence is highly divergent (15% identity and 23% similarity) from its closest characterized relative, CdiA-CT[YKris], and the proteins have different biological roles and target specificities. The bacterial homologs of Ribocin—CdiA-CT[YKris] and CdiA-CT[Kp342]—mediate antagonistic interactions between bacteria as part of systems known as "contact-dependent growth inhibition" (CDI) [42]. The CdiA-CT[YKris] toxin exhibits unspecific RNase activity towards tRNA and rRNAs [30], while CdiA-CT[Kp342] specifically cleaves deacylated $tRNA_{GAU}^{Ile}$ [31]. The mammalian homologs of Ribocin—Amphinase and other RNase A members—include digestive enzymes and participants in stress and antiviral responses [35]. These RNases can be nonspecific (pancreatic RNase A) or specifically cleave tRNAs (angiogenin) [35–37].

To our knowledge, Ribocin—which promotes *P. aeruginosa* virulence in *C. elegans*—is the first bacterial nuclease shown to act by cleaving animals' rRNA. Ribocin's ability to cleave H69 of ribosomes in *C. elegans* or mammalian cells indicates that *P. aeruginosa* could employ Ribocin expression to inhibit translation in a variety of natural animal hosts. Moreover, the finding that Ribocin is active in human cells raises the potential for biotechnological and therapeutic applications.

Ribocin's unique target specificity toward H69 differentiates it from other toxins. Eukaryotic-ribosome-targeting toxic enzymes are structurally unrelated to Ribocin and target distinct sites; these include sarcin (RNase T1 family) [43,44], the ricin/Shiga toxin N-glycosidases [45], and the SidI mannosyltransferase, which glycosylates 25 of the ribosome's proteins [46]. To our knowledge, no known nuclease cleaves bacterial H69, and Ribocin does not either. Ribocin is also structurally unrelated to the bacterial-ribosome-targeting nucleases colicin E3 [5], VapC20 and VapC26 [47,48], RelE1 [49], MazF11 [50] and MazF3 [51].

The mechanism of Ribocin delivery to the worm's cytoplasm is not understood. Ribocin has a signal peptide for periplasmic localization but Ribocin-mediated H69 cleavage is not affected by mutants of distinct *P. aeruginosa* secretion systems [23]. Ribocin delivery may occur through R-bodies [27], which promote H69 cleavage, are endocytosed by worms, and can release cargo proteins by endosomal rupture [43]. Nonetheless, RbcN delivery to the worm cytoplasm occurs also when Ribocin is expressed in *E. coli*, a bacterium without R-bodies, suggesting that other delivery mechanisms are plausible.

Helix 69 is a universally conserved ribosomal element that forms a bridge between the two ribosomal subunits [24] and interacts with the A- and P-sites of the decoding center [24] (Fig 3C). Deletion of bacterial H69 impairs translation [25,26]. Our results provide evidence that H69 cleavage also impairs translation. In addition, Ribocin abolishes translation in mammalian cell lysates with high potency ($IC_{50} = 33\,nM$) comparable to that of angiogenin and other nucleases [36,37,52].

*P. aeruginosa* PA14 virulence toward worms is fully regulated by QS [14]. Downstream of QS, *P. aeruginosa* virulence appears to be multi-factorial, such that multiple individual effectors can function redundantly [13]. Consistent with this model, we propose that PA14 induces *C. elegans* gene expression inhibition through at least three mechanisms. Early during infection (5h), both RbcN and ToxA act redundantly to inhibit translation. However, by 12h, the bacterium elicits host gene expression inhibition via an unknown mechanism that may operate at the levels of transcription and/or translation. Remarkably, the ZIP-2-mediated host response to translation inhibition is predominantly stimulated by Ribocin. Exotoxin A, which is expressed under the tested conditions and can be active against *C. elegans* in certain settings [22], has no apparent contribution to the ZIP-2 host response, in agreement with other reports [21,22]. We speculate that the disparate contributions of ToxA and RbcN to translation inhibition result from differences in activity level and/or their mechanisms of action.

Previous genetic analyses have not identified any virulence effector that is singularly essential for PA14 virulence towards worms. We have confirmed previous reports [22] that ToxA does not contribute to virulence, whereas we show

that RbcN has a measurable contribution to virulence. Altogether, considering the differing effects of ToxA and Ribocin on virulence and the ZIP-2/IRG-1 host response, Ribocin appears to be more consequential. We posit, therefore, that Ribocin, R-bodies—a proximal effector with a measurable impact on virulence [27]—and other factors are the major collective contributors to the virulence of *P. aeruginosa* PA14 toward *C. elegans*.

In conclusion, Ribocin expands the paradigm of bacterial virulence by establishing targeted rRNA cleavage at Helix 69 of eukaryotic ribosomes as a mechanism of host attack. Our findings reveal that Ribocin mediates a ToxA-independent strategy for *P. aeruginosa* to inhibit translation in its eukaryotic hosts. *P. aeruginosa* likely uses these two strategies either jointly or individually to interact antagonistically with distinct eukaryotes. In these biological contexts, the mechanisms mediating the expression, delivery, and modes of action of Ribocin and Exotoxin A may restrict or enable each strategy in their capacity to repress translation.

## Materials and methods

### *C. elegans* strains and genetics

All nematode strains were maintained using standard methods on NGM plates [53] and fed with *E. coli* HB101. The *C. elegans* N2 strain was used as a wild-type strain. The nematode strains used in the present study are listed in S4 Table. A CRISPR/Cas9 genome editing protocol [54] was employed to generate a worm *disl-2* null mutant. Briefly, Cas9-expressing worms were microinjected with two guide RNAs targeting *disl-2* and one guide targeting *dpy-10*. The *dpy* F1 generation was selected and genotyped using PCR. The *dpy-10* mutant was selected against in the subsequent worm generation, while the *disl-2* mutation was maintained. The oligos used to generate the strain are listed in S5 Table.

### Bacterial strains and genetics

All bacterial strains (*P. aeruginosa*, *E. coli*) were routinely grown on Lysogeny Broth (LB) media at 37 °C without antibiotics. The bacterial strains employed in the present study are listed in S6 Table. To generate null mutants of PA14_21120 (named *rbcN*) and *toxA*, we genome-edited *P. aeruginosa* PA14 using its endogenous type I-F CRISPR/Cas3 system, as previously described [55,56]. Briefly, we constructed a gentamicin-selectable "editing plasmid" containing a spacer targeting the *rbcN* or *toxA* gene, along with a homologous recombination template. PA14 bacterial cells were washed twice with 300 mM sucrose and subjected to electroporation with the editing plasmid. Transformants were selected on LB gentamicin plates, re-streaked, and genotyped by PCR. After obtaining the designed genomic modification, the editing plasmid was cured by strain passage in liquid LB culture without antibiotics. The mutant strains were whole-genome sequenced (Plasmidsaurus), along with the parental PA14 wild-type, to verify the absence of additional mutations. The complete *PA14_21120* gene is deleted in Δ*rbcN*. In the Δ*toxA* strain, a premature stop and frameshifting deletion were introduced in the gene. The allele is likely null, as toxA mRNA was undetectable in the strain (S8 Fig). The Δ*rbcN* and Δ*rbcN* Δ*toxA* strains were generated independently from the parental PA14 wildtype. The oligos and plasmids used to generate the bacterial strains are listed in S5 Table.

### *C. elegans*—*P. aeruginosa* interaction assays

Exposure of *C. elegans* to *P. aeruginosa* was performed using SK conditions [12]. In summary, an aliquot from an overnight liquid LB culture of *P. aeruginosa* was spread across an SK agar plate to cover the entire surface of the agar, thus preventing worms from easily escaping the bacterial lawn. The plates were incubated first at 37 °C for 24 h and then at 25 °C for 24 h to allow lawn growth and the induction of bacterial pathogenic activity [12]. In parallel, a synchronous population of *C. elegans* worms was prepared by standard hypochlorite treatment, followed by culture of larvae until the young adult stage on NGM agar plates seeded with *E. coli* HB101. The young adult worms were then transferred to the SK plates to initiate their exposure to *P. aeruginosa*.

## Ribosome cleavage assays

Purified, in vitro reconstituted ribosome complexes were prepared as follows. *Oryctolagus cuniculus* (European rabbit) ribosomal subunits were isolated from rabbit reticulocyte lysates not treated by micrococcal nuclease (RRL; Green Hectares), and bacterial ribosomal subunits were isolated from *E. coli* lysates, as described previously, using sucrose gradient ultracentrifugation and fractionation [57]. Components of mammalian 80S or bacterial 70S ribosomal complexes were then assembled as described [36,58] to final concentrations of 0.5 μM 40S, 20 μM mRNA, 1.2 μM tRNA$^{fMet}$ (Chemical Block), and 0.5 μM 60S (for 80S ribosome) or 2.5 μM 30S, 12 μM mRNA, 5 μM tRNA$^{fMet}$, and 2.5 μM 70S. Complexes were stored at −80 °C until use.

In vitro H69 cleavage reactions contained a Ribocin source (fractionated S100 lysate or recombinant RbcN) and substrate ribosomes, mixed in the reaction buffer (50 mM potassium acetate, 20 mM Tris-acetate, 10 mM magnesium acetate, and murine RNase inhibitor at 1 U/μL). A typical assay was incubated at 26 °C for 20 min, ice-quenched, and subjected to RNA isolation (guanidinium thiocyanate, phenol/chloroform, and isopropanol precipitation). RNA analysis was performed using capillary electrophoresis (described in the section Analysis of ribosomal RNA profiles and fragment quantification). For the assays using 70S ribosomes, side-by-side reactions using rabbit ribosomes served as positive controls of catalytically active Ribocin.

## Analysis of ribosomal RNA profiles and fragment quantification

To extract RNA from any sample type (e.g., worms, 80S rabbit ribosomes, RRL), the sample was treated with acid guanidinium thiocyanate (QIAzol reagent, Qiagen) and subjected to RNA isolation (phenol/chloroform and isopropanol precipitation). The obtained RNA was quantified, and capillary electrophoresis was performed using a 5300 Fragment Analyzer system (Agilent). Electropherograms were analyzed using the ProSize 2.0 software (Agilent) to quantify the distinct rRNA species, including H69-cleaved fragments.

## Cloning of RNA termini

To determine the 5′ termini of rRNA samples, a phosphorylated linker RNA was ligated using T4 RNA ligase 1. Ligation was followed by random hexamer-primed cDNA synthesis using Superscript III reverse transcriptase (Thermo Fisher Scientific, Massachusetts, USA). PCR amplification was carried out with primers complementary to the linker and multiple positions along the 26S rRNA. The obtained amplicons were then either Sanger sequenced directly or after TOPO cloning.

## Biochemical fractionation of H69 nuclease activity

***Preparation of S100 lysates.*** Synchronized adult *C. elegans* were exposed to PA14 on SK plates for 24 hours, collected, washed with M9 buffer, and pelleted. After lysis using a Dounce grinder and complete lysis buffer (30 mM HEPES pH 7.4, 2 mM MgCl$_2$, 50 mM KCl, 0.1% Triton X-100, 5% glycerol, 2.5 mM DTT, 1 mM PMSF, plus an EDTA-free protease inhibitor tablet (Roche), and SUPERase-In RNase inhibitor (Thermo Fisher Scientific)), worm lysates were cleared by centrifugation (5 min at 3,300 rpm followed by 20 min at 12,700 rpm) in an Eppendorf 5430R centrifuge. The cleared lysates underwent ultracentrifugation (1 hour at 55,000 rpm using an Optima MAX-TL centrifuge equipped with a TLA-100 rotor), and the obtained supernatant was retained as the S100 lysate. A control S100 extract was prepared from synchronized worms maintained on non-pathogenic *E. coli* HB101.

**Activity-based fractionations.** All the fractionation approaches relied on the H69-cleaving activity present in S100 lysates from worms exposed to *P. aeruginosa* PA14. The lysates were subjected to a fractionation procedure described below, and fractions were tested for activity using cleavage assays (described in the section Ribosome cleavage assays). In these cleavage assays, rabbit 80S ribosomes served as the substrate, without added mRNA or tRNA. The nuclease

activity present in S100 lysates can cleave rabbit 80S ribosomes at H69 (S9 Fig), just as it cleaves *C. elegans* ribosomes at H69 [23]. In a standard assay reaction, aliquots of the collected fractions were equalized by protein mass and tested on approximately 250 fmol of 80S ribosomes. Following quantification of the rRNA fragments resulting from H69 cleavage, we calculated the nuclease activity in terms of specific activity (i.e., femtomols of H69-cleaved 80S per minute per µg of lysate) and total units of activity per fraction.

Each fractionation procedure consisted of several steps carried out in tandem. After each fractionation step, we quantified nuclease activity (specific activity and total units), and the fractions encompassing the top peak activity were merged and selected for further fractionation. In the first tandem fractionation procedure, an S100 lysate from PA14-exposed N2 worms was fractionated by cation exchange chromatography using a HiPrep SP 16/10 column (Cytiva), followed by a HiTrap Heparin HP column (Cytiva), and then by size-exclusion chromatography (Sephadex 10/300 column). The chromatographic separations were conducted using an AKTA Explorer FPLC (GE Healthcare) system. Upon final fractionation, three fractions (named S6–S8) were selected for protein identification by mass spectrometry.

In the second tandem fractionation procedure, *glp-1(ts)* worms were raised at 25 °C (the restrictive temperature that blocks growth of the germline) and exposed to either PA14 or *E. coli* HB101 (control lysate condition). The worms were processed into two separate S100 lysates (described in "Preparation of S100 lysates"), which were then fractionated in parallel. The first lysate fractionation employed ammonium sulfate (AS) precipitation in 5% AS increments and was followed by cation exchange chromatography using a SP column (Cytiva). The two most enriched fractions for H69-cleaving activity (named IF and IW) were selected for protein identification by mass spectrometry, along with control lysate fractions "UF" and "UW," which correspond methodologically to fractions IF and IW.

**Sample processing for mass spectrometry.** Twenty-five µg of each sample was mixed with digestion buffer (8 M urea, 0.1 M Tris-HCl, pH 8.5). The samples were then reduced and alkylated through sequential 20-min incubations with 5 mM TCEP (Tris 2-carboxyethyl phosphine) and 10 mM iodoacetamide at room temperature in the dark, while being mixed at 1,200 rpm in an Eppendorf ThermoMixer. Twelve µl of carboxylate-modified magnetic beads (CMMB, also widely known as SP3) [59] was added to each sample. Ethanol was added to a concentration of 50% to induce protein binding to the CMMB. The CMMB were washed three times with 80% ethanol and then resuspended in 50 µl of 50 mM TEAB. The protein was digested overnight with 0.1 µg LysC (Promega) and 0.8 µg trypsin (Thermo Scientific, 90057) at 37 °C. Following digestion, 1.2 ml of 100% acetonitrile was added to each sample to increase the final acetonitrile concentration to over 95%, facilitating peptide binding to the CMMB. The CMMB were then washed three times with 100% acetonitrile, and the peptide was eluted with 65 µl of 2% DMSO. Eluted peptide samples were dried by vacuum centrifugation and reconstituted in 5% formic acid before analysis by LC–MS/MS.

**LC–MS acquisition and analysis.** Peptide samples were separated on a 75 µM ID, 25 cm C18 column packed with 1.9 µM C18 particles (Dr. Maisch GmbH) using a 140-min gradient of increasing acetonitrile concentration and injected into a Thermo Orbitrap Fusion Lumos Tribrid mass spectrometer. MS/MS spectra were acquired using Data Dependent Acquisition (DDA) mode. MS/MS database searching was performed using MaxQuant [60], version 1.6.10.43 against the *C. elegans* reference proteome from WormBase and the *Pseudomonas aeruginosa* strain PA14 reference proteome from *Pseudomonas* Genome DB. The mass spectrometry features of the proteins in the fractionated samples are listed in S7 Table.

**Candidate ranking.** At the end of each tandem fractionation procedure (TFP), multiple closely related fractions were selected to capture: 1—the peak H69-cleaving activity, 2—a sample with less activity than the peak, and 3—a sample with no activity. In the first TFP, fraction S6 had no activity, fraction S7 had peak activity, and fraction S8 had intermediate activity. In the second TFP, fraction IW had peak activity, fractions IF and IF2 had intermediate activity, and fractions UF and UW had no activity.

The proteins detected (with a peptide count >0) in all active fractions (S7, S8, IW, IF) were selected for score ranking. Two weighted scores were assigned per protein: the "absence score" ($S_A$) and the "pattern score" ($S_P$). To calculate the

scores, the "peptide count" values of a given protein are indicated by "P" followed by the underscored fraction name (i.e., $P_{S6}$ to $P_{S8}$, $P_{IF}$, $P_{IW}$, $P_{UF}$, $P_{UW}$). The "absence score" assigns 1 point when each of the following equations holds true: $P_{S6} = 0$, $P_{UF} = 0$, $P_{UW} = 0$. The score varies from 0 to 3. The "pattern score" awards 3 points when each of the following equations holds true: $P_{S7} > P_{S6}$, $P_{S7} > P_{S8}$, $P_{IW} > P_{IF}$, $P_{IW} > P_{IF2}$, $P_{IW} > P_{UW}$, $P_{IF} > P_{UF}$. The score varies from 0 to 18. A protein candidate with perfect fit to the activity expectations has $S_A = 3$ and $S_P = 18$. To rank the candidates based on the two scores, the score variables were normalized (0–1) and the taxicab distance ($\delta$) to $S_A = 3$, $S_P = 18$ was calculated, as follows: $\delta = (1/3) \times |S_A - 3| + (1/18) \times |S_P - 18|$. The obtained scores for the protein candidates and their ranking are presented in S1 Table. The presented candidate scoring and ranking system is a novel experimental design feature for fractionation experiments that may be useful in other activity-based approaches to identify biological functions.

### Protein sequence and structure analyses

Ribocin homologs were identified using BLASTp and the NCBI ClusteredNR database. The obtained homolog sequences were inspected for domain annotations using NCBI's Conserved Domain Database. Multiple sequence alignment was carried out using Clustal Omega [61] and the ESPript 3 server [62]. Phylogenetic analysis was performed using "Simple Phylogeny" [63,64]. The predicted structure of the RbcN protein was obtained from the AlphaFold database version 2 [28]. The search for RbcN homologs in the Protein Data Bank (PDB) and structural alignments were performed using the protein structure comparison Dali server [29].

### Ribocin expression in *E. coli*

The full-length coding sequence (cds) of the *P. aeruginosa* PA14 *rbcN* gene was cloned into pET28a expression plasmids. Two plasmids were constructed: one with the wild-type *rbcN* cds and the other with an H60A point mutant (where histidine 60 is replaced with alanine). The plasmids were transformed into *E. coli* BL21, and bacterial cultures were grown in LB medium to ~0.4 OD$_{600}$, then either induced (with IPTG) or maintained as uninduced control cultures (no IPTG added). The bacteria were then plated on NGM plates. Synchronized young adult worms were then placed on the plates to initiate the experiments. The designed plasmids were synthesized by Twist Bio and are listed in S5 Table.

### Production of recombinant Ribocin

The coding sequence (cds) of the *P. aeruginosa* PA14 *rbcN* gene was cloned into pET28a expression plasmids. In the constructs, the *rbcN* signal peptide was removed, and an N-terminal 6x-His tag was incorporated. Two plasmids were created: one containing the wild-type *rbcN* cds and the other harboring the H60A point mutant (histidine 60 replaced with alanine). The designed plasmids were synthesized by Twist Bio and are listed in S5 Table. These plasmids were transformed into *E. coli* BL21 cells. Bacterial cultures were grown to an optical density of 0.8 and induced with a final IPTG of 1 mM. Cells were resuspended in lysis buffer [50 mM Tris (pH 8), 2 mM EDTA (pH 8)] and lysed by sonication using 5 cycles of 40 s bursts at 50% amplitude. Bacterial inclusion bodies were isolated by centrifugation at 20,000 *g* for 20 min at 4 °C before resuspending in 10 mL of solubilization buffer containing 100 mM Tris (pH 8), 2 mM EDTA (pH 8), 7 M guanidinium hydrochloride, and 150 mM reduced L-glutathione. Solubilization proceeded with stirring under inert atmosphere (argon) for 2 hours at room temperature. The protein was refolded under standard oxidative refolding conditions by diluting dropwise into 500 mL of (0.6 mM) oxidized L-glutathione and 500 mM L-arginine (pH 8) [65]. The refolded recombinant RbcN was clarified by centrifugation at 10,000 *g* for 20 min then subjected to a Ni-NTA column (Cytiva Life Sciences). Bound protein was washed with 10 column volumes (CV) of wash buffer [50 mM Tris (pH 8), 100 mM NaCl, 10 mM Imidazole] and eluted in 2 CV of wash buffer supplemented with 200 mM Imidazole. Recombinant RbcN was buffer exchanged into 50 mM HEPES pH 7.5 with 50 mM KCl and stored at −80 °C.

## In vitro translation assays

In vitro translation assays in rabbit reticulocyte lysates (Green Hectares) were conducted as previously described [34] with the modification that rabbit reticulocyte lysates not treated with micrococcal nuclease (Green Hectares) were used, prepared as in [36]. Briefly, translation mixtures containing 1% nanoluciferase substrate furimazine (Promega) were pre-incubated with RbcN or a control buffer for 5 min at 30 °C. Then, nanoluciferase reporter mRNA was added to 30 nM and luminescence signal was recorded in kinetic mode using a Tecan Spark instrument. For bacterial translation, translation-competent *E. coli* extract (NEBExpress, New England Biolabs) was prepared according to manufacturer recommendations without addition of T7 polymerase as in [57]. RRL assay conditions were replicated in this *E. coli* extract with the modification that the reporter mRNA contained a Shine-Dalgarno sequence upstream of the nanoluciferase coding region. To estimate the $IC_{50}$ of Ribocin, the maximal rate of translation for the in vitro translation assays was calculated and a three-parameter log-logistic model was fitted using the *drc* package (version 3.0) in R [66]. After 30 min, all translation reactions were terminated by addition of TRIzol (Invitrogen). Total RNA was extracted and subjected to capillary electrophoresis as described.

## Electroporation of recombinant RbcN protein into human cells

Human HEK-293T (293T) cells were maintained in DMEM medium supplemented with 10% fetal bovine serum (FBS) and 1% penicillin/streptomycin (Pen-strep). The mammalian cells were electroporated using a Neon instrument system (Thermo Fisher) using a 10 μL tip kit and following the manufacturer's protocol. The 293T cells were transfected in 1 μM recombinant RbcN solution and the following parameters: 1,150 V, 20 ms and 2 pulses.

## Luciferase assays in vivo

To assess the effect of RbcN on translation in vivo, 293T cells were electroporated with nanoluciferase mRNA and RbcN. Electroporated cells (~40,000/well) were plated in a 96-well plate (Corning) with 100 μL of DMEM supplemented with 10% FBS, without antibiotics. Three hours post-transfection, the media was replaced with 25 μL Opti-MEM + 1% nanoluciferase substrate endurazin (Promega), and luminescence recorded using a Tecan Spark instrument.

## Puromycin incorporation assays

To measure translation in *C. elegans* worms a procedure was derived from the SunSET method [67]. A fixed number of adult worms were put on SK plates with a *P. aeruginosa* strain lawn for a designated time (e.g., 5 or 12 h), collected and incubated for 1 hour in a second SK plate carrying the same bacterial strain and 5 mM puromycin (Santa Cruz Biotech). The worms sample were then washed with M9 buffer, freeze-thawed with liquid nitrogen, and boiled in Laemmli buffer. Total protein was quantified on a Typhoon FLA 9500 instrument (GE Healthcare Life Sciences) using the TotalStain Q PVDF fluorescent total protein staining kit (Azure Biosystems). The worm samples were separated by electrophoresis on 4%–20% gels and blots developed with primary monoclonal anti-puromycin antibody (12D10, Sigma, RRID: AB_2566826) or monoclonal anti-tubulin (T6074, Sigma) and goat anti-mouse HRP-conjugated secondary antibody (1721011, Biorad, RRID: AB_11125936) secondary antibody. The blot chemiluminescence was measured using the SuperSignal West Femto Maximum Sensitivity Substrate (34095, ThermoFisher) and the Amersham Imager 600 instrument (GE Healthcare Life Sciences). The mean signal intensity for puromycin was measured using Fiji ImageJ software (version 1.54f) and normalized to that of tubulin.

## Western blot analyses

To measure the abundance of the RbcN protein, a polyclonal anti-RbcN antibody (α-RbcN, Sino Biological) was developed. Purified recombinant RbcN (described in the section Production of recombinant Ribocin) was used as the antigen.

Three rabbits were immunized, and the resulting serum was affinity-purified using protein A and RbcN. For Western blots, the bacterial samples were separated by electrophoresis on 10%–20% tricine gels (Novex, EC6625BOX) and blots were developed with α-RbcN serum (1:500 dilution) and HRP-conjugated donkey anti-rabbit antibody (NA934, Cytiva, RRID: AB_772206) secondary antibody. Chemiluminescence and image analysis was conducted as for the "Puromycin incorporation assays." Western membranes were stained for total protein with a fluorescent dye (TotalStain Q - PVDF kit, Azure Biosystems) following the manufacturer's protocol and imaged using a Typhoon FLA 9500 instrument (GE Healthcare Life Sciences).

## Real-time quantitative PCR

To measure RNA abundance in bacterial and worm samples, total RNA was isolated (guanidinium thiocyanate, phenol/chloroform, and isopropanol precipitation). Complementary DNA (cDNA) was synthesized by reverse transcription (Protoscript) using random primers hexamers (ThermoFischer). Real-time quantitative PCR (qPCR) was performed using a SYBR green mixture (FastSYBR mixture, CWBio) in a QuantStudio 3 instrument (Thermo). Sequences of the primers used are found in S5 Table. For the qPCRs with *C. elegans* samples, six reference genes (*act-1, tba-1, pmp-3, idhg-1, mdh-1, iscu-1*) that have been previously proposed as reference genes [68,69] were evaluated using the samples and the geNorm algorithm [70]; *act-1* was selected as the most reliable reference gene. For the qPCRs with *P. aeruginosa* samples, the *clpX (PA14_41230)* gene was used as reference gene [71]. In all cases, relative gene expression was calculated using the ΔΔCt method.

## Worm survival analysis

Worms were exposed to *P. aeruginosa* under SK conditions (described in the section *C. elegans—P. aeruginosa* interaction assays). The time course of worm death was manually determined by prodding the worms with a wire pick. The collected survival data was analyzed using R (Survival package version 3.5 and Survminer package version 0.4.9) with the Kaplan–Meier (K-M) method. Statistical comparisons between survival curves were done using the log-rank test and Tarone–Ware test (rho = 0.5) using Survminer. The survival assays were performed with three or more independent biological replicates, and the results were consistent across all independent replicate experiments.

## Microscopy

Transgene fluorescence (GFP or mCherry) was examined using a Zeiss Imager Z1 microscope, an Axiocam 503 camera, and Zen Blue software. Transgenic worms (*irg-1::GFP*) were grown on *E. coli* HB101 at 20 °C, synchronized by hypochlorite treatment, and exposed to *P. aeruginosa* strains at the young adult stage. Worm images were acquired using a 10× objective and constant exposure settings (200 ms). The images were further processed and quantified using Fiji/ImageJ [72,73].

## Statistical analysis

All experiments were performed with at least two biological replicates. All the statistic methods were conducted using the R language (version 4.2.2) and associated software packages. All the performed t-tests were two-tailed. The plotted figures were prepared using R and the ggplot2 software package (version 3.4.1).

## Supporting information

**S1 Fig. Features and Conservation of the Ribocin (RbcN) protein. (A)** Protein sequence alignment of PA14_21120 and its ortholog PA3318 in *P. aeruginosa* strain PA01. The signal peptide sequence is underlined. **(B)** Unrooted phylogenetic tree of Ribocin homologs. Each node indicates the name of a bacterial taxon carrying an RbcN homolog. Arrow

indicates *P. aeruginosa* Ribocin. The data underlying this panel can be found in <u>S2 Data</u>. **(C)** AlphaFold-predicted structure of RbcN (PA14_21120), colored by pLDDT score. The unstructured signal peptide is colored yellow. **(D)** Location in the AlphaFold RbcN structure of four conserved amino acids. H60 and K75 in red, T159 in green, and Y161 in blue color.
(TIF)

**S2 Fig. Alignment of Ribocin (RbcN) and sequence homologs.** Protein sequence alignment of *P. aeruginosa* RbcN and the sequence homologs shown in <u>S1B Fig</u>. Identical or similar amino acids are shown in red font and boxed in blue.
(TIF)

**S3 Fig. Recombinant RbcN protein purity and activity. (A, B)** Coomassie-stained gels of the recombinant RbcN isolation procedure for wild-type (wt) (A) and H60A mutant (B) protein. Following refolding, RbcN-containing samples were passed through a Ni-NTA column. The protein profiles of flow-through, wash, and eluate samples are shown. The full-length RbcN protein is indicated by arrows. L, protein ladder. kDa, kilodalton. **(C)** Denaturing agarose gel for an mRNA assayed for cleavage by recombinant RbcN (wt or H60A). Arrowheads indicate RNA ladder markers ($\times 10^3$ nucleotides).
(TIF)

**S4 Fig. Effect of Ribocin on prokaryotic ribosomes. (A, B)** RNA profiles of *Escherichia coli* ribosomes in translationally-competent 30S lysate after treatment with recombinant RbcN wild-type (wt) or H60A mutant RbcN at the indicated concentrations. The individual RNA profiles are overlayed in (A). "M" indicates a 15-nucleotide (nt) marker. R.F.U. represents relative fluorescence units.
(TIF)

**S5 Fig. Effects of *rbcN* on translation and mRNA abundance. (A)** Quantification of the mScarlet-I3::PEST fluorescence upon worm exposure to cycloheximide (CHX) for 3.5 hours (h). **(B)** Relative abundance of the *mScarlet-I3::pest* mRNA by RT-qPCR for the conditions shown in (A). **(C)** Fluorescence microscopy images of *mScarlet-I3::pest* worms (strain VT4414) exposed to *E. coli* expressing *rbcN* wild-type (wt) or H60A for 3 hours. Scale bar measures 50 μm for all images. **(D)** Quantification of the mScarlet-I3::PEST mean fluorescence intensity for worms exposed to *E. coli* expressing *rbcN* wt for 7 or 12 h. **(E)** Relative abundance of the *mScarlet-I3::pest* mRNA and pre-mRNA measured by RT-qPCR for the 7 h condition shown in (D). **(F)** Relative mRNA abundance of four intestine-specific genes (*ges-1, ifb-2, vha-6, vit-6*) by RT-qPCR for the 7 h conditions shown in (D). The intermediate filament *ifb-2* is induced by host responses. The data underlying this figure can be found in <u>S1 Data</u>. "n.s." denotes not significantly different. "\*\*\*" indicates a *p*-value < 0.001 for two-sided Welch *t* test comparisons.
(TIF)

**S6 Fig. Effect of *rbcN* and *toxA* on bacterial virulence. (A–D)** Survival curves of adult worms exposed to PA14 (wt, in red) or Δ*rbcN*. (A,C); Δ*toxA* (B); or Δ*rbcN* Δ*toxA* (D). The *p*-value from a log-rank test curve comparison and *n* values (individual animals) are shown. The data underlying this figure can be found in <u>S1 Data</u>.
(TIF)

**S7 Fig. Effect of *rbcN* on the host ZIP-2/IRG-1 pathway.** Fluorescence microscopy images of *irg-1::GFP* worms exposed to *E. coli* heterologously expressing the *rbcN* gene, either wild-type (wt) or H60A mutant. Scale bar measures 50 μm for all images.
(TIF)

**S8 Fig. Characterization of the Δ*toxA* strain.** RT-qPCR of relative *toxA* mRNA abundance in the Δ*toxA* strain and PA14 wt. "\*\*" indicates a *p*-value < 0.01 for a two-sided Welch *t* test comparison. The data underlying this figure can be found in <u>S1 Data</u>.
(TIF)

**S9 Fig. H69 cleavage of rabbit ribosomes in vitro. (A)** Total RNA profiles of 80S rabbit ribosomes treated with S100 lysate from PA14-exposed worms or mock-treated. Arrows indicate the H69-cleaved fragments with their estimated nucleotide size ($\times 10^3$). **(B)** Sanger sequencing trace indicating the cloned cleavage site (vertical line) from rRNA of S100-treated rabbit ribosomes shown in (A). **(C)** Cleavage site (arrowhead) indicated in the Helix 69 rRNA secondary structure. "M" denotes a 15-nt marker. R.F.U. stands for relative fluorescence units; nt, nucleotide; H69, helix 69.
(TIF)

**S1 Table. Scores and ranking of H69 nuclease protein candidates.**
(XLSX)

**S2 Table. Taxonomic classification of species with RbcN homologs.**
(XLSX)

**S3 Table. Structural Ribocin homologs identified using the Dali server.**
(XLSX)

**S4 Table. *Caenorhabditis elegans* strains used in the present study.**
(XLSX)

**S5 Table. Oligonucleotides and plasmids used in the present study.**
(XLSX)

**S6 Table. Bacterial strains used in the present study.**
(XLSX)

**S7 Table. Proteins identified by mass spectrometry analysis of fractionated samples.**
(XLSX)

**S1 Data. Spreadsheet containing the data points used for all plotted graphs in the manuscript (Figs 1–5, S4–S6, S8, and S9).**
(XLSX)

**S2 Data. Phylogenetic tree file for the RbcN homologs (S1B Fig).**
(TREE)

**S1 Raw Images. Uncropped images for Figs 2 and S3.** Reference to main figure panels is made in the figure. Please refer to the respective legend.
(PDF)

## Acknowledgments

We would like to acknowledge members of the Ambros, Korostelev and Mello laboratories for their feedback on this research project. We acknowledge the Mitani lab as the source of multiple tested *Caenorhabditis elegans* strains. Some of the investigated *C. elegans* strains were provided by the CGC, which is funded by the NIH Office of Research Infrastructure Programs (P40 OD010440).

## Author contributions

**Conceptualization:** Alejandro Vasquez-Rifo, Denis Susorov, Emily H. Sholi, Andrei Korostelev, Victor Ambros.

**Data curation:** Alejandro Vasquez-Rifo, Denis Susorov, Emily H. Sholi.

**Formal analysis:** Alejandro Vasquez-Rifo, Denis Susorov, Emily H. Sholi, Gabriel Demo.

**Funding acquisition:** Alejandro Vasquez-Rifo, James A. Wohlschlegel, Andrei Korostelev, Victor Ambros.

**Investigation:** Alejandro Vasquez-Rifo, Denis Susorov, Emily H. Sholi, Yasaman Jami, Jihui Sha, James A. Wohlschlegel.

**Methodology:** Alejandro Vasquez-Rifo, Denis Susorov, Emily H. Sholi, Gabriel Demo, Yasaman Jami, Jihui Sha, James A. Wohlschlegel, Andrei Korostelev, Victor Ambros.

**Project administration:** Andrei Korostelev, Victor Ambros.

**Resources:** Gabriel Demo.

**Software:** Alejandro Vasquez-Rifo.

**Supervision:** Alejandro Vasquez-Rifo, James A. Wohlschlegel, Andrei Korostelev, Victor Ambros.

**Validation:** Alejandro Vasquez-Rifo, Denis Susorov, Emily H. Sholi.

**Visualization:** Alejandro Vasquez-Rifo, Emily H. Sholi, Andrei Korostelev.

**Writing – original draft:** Alejandro Vasquez-Rifo, Andrei Korostelev, Victor Ambros.

**Writing – review & editing:** Alejandro Vasquez-Rifo, Denis Susorov, Emily H. Sholi, Gabriel Demo, Yasaman Jami, Jihui Sha, James A. Wohlschlegel, Andrei Korostelev, Victor Ambros.

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
