## [Editor Report · Decision Letter 0]

4 Jan 2026

Dear Dr Vasquez-Rifo,

Thank you for submitting your manuscript entitled "P. aeruginosa Ribocin cleaves ribosomal Helix 69 to inhibit host translation" for consideration as a Research Article by PLOS Biology. Please accept my apologies for the long delay.

Your manuscript has now been evaluated by the PLOS Biology editorial staff, as well as by an academic editor with relevant expertise, and I am writing to let you know that we would like to send your submission out for external peer review as an Update Article, considering your previous publication with us.

However, before we can send your manuscript to reviewers, we need you to complete your submission by providing the metadata that is required for full assessment. To this end, please login to Editorial Manager where you will find the paper in the 'Submissions Needing Revisions' folder on your homepage. Please click 'Revise Submission' from the Action Links and complete all additional questions in the submission questionnaire. Please Select "Update Article" in the article type.

Once your full submission is complete, your paper will undergo a series of checks in preparation for peer review. After your manuscript has passed the checks it will be sent out for review. To provide the metadata for your submission, please Login to Editorial Manager (https://www.editorialmanager.com/pbiology) within two working days, i.e. by Jan 07 2026 11:59PM.

Kind regards,

Melissa

Melissa Vazquez Hernandez, Ph.D.

Associate Editor

PLOS Biology

---

## [Decision Letter · Decision Letter 1]

28 Jan 2026

Dear Dr Vasquez-Rifo,

Thank you for your patience while your manuscript "P. aeruginosa Ribocin cleaves ribosomal Helix 69 to inhibit host translation" went through peer-review at PLOS Biology. Your manuscript has now been evaluated by the PLOS Biology editors, an Academic Editor with relevant expertise, and by several independent reviewers.

In light of the reviews, which you will find at the end of this email, we are pleased to offer you the opportunity to address the comments from the reviewers in a revision that we anticipate should not take you very long. We will then assess your revised manuscript and your response to the reviewers' comments with our Academic Editor aiming to avoid further rounds of peer-review, although we might need to consult with the reviewers, depending on the nature of the revisions.

As you will see in the reports, all reviewers are quite positive about your work but there are still some experimental concerns that must be addressed. Reviewer 1 raises a methodological concern about the lack of biological replicates in worm survival assays, along with a few minor figure-related suggestions. Reviewer 2 asks for a negative control on the survival assays, and encourages further discussion of Ribocin’s substrate specificity. Reviewer 3 suggests expanding the discussion of Ribocin secretion and uptake to better contextualize host–pathogen physiology. Reviewer 4 raises only minor requests to broaden the discussion of ribosomal damage and specificity, particularly regarding eukaryotic versus bacterial ribosomes. We agree with all the reviewers' concerns and suggestions, and we will require to beto be addressed for further consideration.

We expect to receive your revised manuscript within 2 months. Please email us (plosbiology@plos.org) if you have any questions or concerns, or would like to request an extension.

**IMPORTANT - SUBMITTING YOUR REVISION**

*Resubmission Checklist*

*Published Peer Review*

*PLOS Data Policy*

*Blot and Gel Data Policy*

Sincerely,

Melissa

Melissa Vazquez Hernandez, Ph.D.

Associate Editor

PLOS Biology

REVIEWERS' COMMENTS

Reviewer #1:

In the manuscript, "P. aeruginosa Ribocin cleaves ribosomal Helix 69 to inhibit host translation", the authors find and characterize a new ribonuclease, Ribocin, showing that it is responsible for the helix 69 cleavage caused by P. aeruginosa that they previously discovered. The study begins by screening infected worm lysates for the H69 cleavage activity using a biochemical approach. Based on the list of candidate proteins, they further screened publicly available C. elegans and P. aeruginosa mutants and found one PA14 mutant in which activity was completely lost. The mutation was in an uncharacterized gene, but with the help of AlphaFold, it was found to resemble proteins in the RNase A superfamily. They then created point mutants in conserved residues crucial for RNase activity and demonstrated a loss of activity in both in vivo (C. elegans lysates) and in vitro (recombinant protein) settings. They also expressed the protein in heterologous systems (E. coli and eukaryotic cells) and demonstrated that it was sufficient to endow the H69 cleavage activity. They also showed that Ribocin's translation-inhibition activity occurs in the presence of this enzyme by adding it recombinantly to an RRL assay and by using in vivo approaches in both human cells and C. elegans. The robustness of the results is bolstered by the use of two complementary methodologies to assess translational inhibition, along with inactive and mutant enzyme controls. Finally, the authors demonstrated that in the worm infection model, where the activity was initially discovered, the Ribocin gene is essential for the translation-blocking activity and is more important than ToxA, a toxin with similar translation-blocking properties. They also observed that loss of the gene results in slightly less PA14 killing of C. elegans. Overall, I find the characterization of a new prokaryotic ribonuclease, with a novel mechanism of acting on eukaryotic cells, to be a fascinating and important contribution. The study is thorough and convincing, with results supported by multiple approaches and rigorously conducted. My only major concern was with how the survival assays were carried out, and is described below.

Major Comments

1) Survival Assays: Due to the inherent variability associated with these assays, it is common practice in the field to include multiple biological replicates, typically three. This was not done in this study. Rather, single assays are shown with a large n (150-200). The large sample size was probably necessary to detect the very small effect of the loss of the rbcN gene on survival. However, I think it is really important to have multiple biological replicates to further probe the influence of this one gene on survival and to avoid overinterpretation. It will not, in my opinion, detract from the impact of this manuscript if the effect proves inconsistent across multiple replicates. Pseudomonas possesses numerous virulence factors, and it is logical that losing just one may be insufficient to affect host survival, a complex, multifactorial phenotype.

Minor Comments

1) Figures 4I and J would be further clarified by listing the time point in the panels, so that the reader does not need to refer to the legend to understand their difference. In fact, they could be combined into a single panel with both time points displayed together.

2) Figure 4 shows all the conditions using both the mScarlet reporter strain and the puromycin assay, except for the PA14 mutants shown in Figures 4I and 4J. Was there a reason the mScarlet assay was not additionally used in this experiment? If the authors have it, could it be included here or put in the supplement?

Reviewer #2:

In this work, Ambros and coworkers identify and characterize Ribocin/RbcN, a nuclease of P. aeruginosa that specifically cleaves H69 of eukaryotic ribosomes. They show that RbcN is distally related to CdiA (another bacterial endonuclease toxin) and is sufficient for H69 cleavage and translation inhibition in C. elegans. They purified the protein and showed that it inhibits translation in vitro (rabbit reticulysate), with an IC50 value of 33 nM. They also provide evidence that RbcN inhibits translation in human and worm cells. In the worm, this activity is revealed in the absence of ToxA, at early (5 h) exposure time. Finally, the authors show that RbcN contributes to P. aeruginosa virulence in the worm and is critical for host IRG-1 response pathway. Overall, this is nice work, and I have only a few recommendations for the paper.

Specific comments:

RbcN cleaves H69 of various eukaryotic ribosomes. Does it act on extracted rRNA? Unmodified (transcript) rRNA? Does RbcN also target H69 of archaea and/or bacteria? Some further discussion of the enzyme's specificity would be appropriate, even if the questions remain open.

Fig. 5A shows the survival rates of worms exposed to PA14 and the del-rbcN strain. Missing is an appropriate negative control (i.e., avirulent P. aeruginosa or E. coli).

In Fig. 4I-J, the data are "tubulin-normalized," based on the legend. I presume this means that the data of I and J are directly comparable, despite that the Y units are arbitrary. If so, there is a dramatic drop in puro-peptide signal across the board, for even the avirulent strain. Why would this be? Perhaps I am missing something, but regardless, this section of the manuscript should be clarified.

Fig. 3B, right panel. The three traces seem intentionally offset via total signal, to see the three colors. This is a bit confusing, especially when different concentrations (albeit of protein) are listed in the key. I would suggest that all four traces (including wt, left panel) be normalized to total signal and then positionally offset (vertically), so the effects are clearer.

Check callouts to Fig. 3B-C (page 9, second paragraph). It seems like panel B, right and left, is what is meant?

Typo: Fig. 2 legend, "starts point to unspecified bands."

Reviewer #3:

The manuscript by Vasquez-Rifo et al. describes the characterization of a Pseudomonas aeruginosa ribonuclease, Ribocin, that carries out the cleavage of the large ribosomal RNA of the C. elegans host. This is an important and rigorous follow-up study to the authors' previous observation that the pathogen P. aeruginosa can inhibit C. elegans host translation by cleavage of the large ribosomal RNA at helix 69. The study is logically presented and experiments establish that Ribocin is necessary and sufficient even when heterologously expressed in E. coli, to carry out the translational inhibition by cleaving the large ribosomal RNA of C. elegans. I only have one comment that may help the audience better appreciate the physiology of the observed interaction—can the authors discuss further whether Ribocin is secreted by bacteria, and if so, how might it be taken up by the intestinal cells of C. elegans, as P. aeruginosa itself is thought to remain largely in the intestinal lumen during infection.

Reviewer #4:

In this manuscript, Vasquez-Rifo et al. carried out comprehensive characterization of P. aeruginosa H69 nuclease, which is named Ribocin. Specifically, the authors identified Ribocin in vivo and performed ribosomal RNA cleavages both in vivo and in vitro, demonstrating that ribosomes from various eukaryotic organisms are the effective substrates of Ribocin. In addition, the authors demonstrated that Ribocin inhibits eukaryotic protein translation in vivo, and it contributes to the virulence of the encoded bacteria. The experimental data provided are abundant and of high quality. This reviewer particularly appreciated the fact that, in addition to ribotoxins targeting sarcin-ricin loop (SRL) and the decoding center of the ribosome, a new ribotoxin specifically targeting H69 has been discovered and comprehensive characterized. Therefore, the study described in the manuscript is highly significant.

This reviewer has no major issues with the manuscript. However, the manuscript could be improved if the authors can address the following two minor issues. Those two issues are to address ribosomal damage in a broader context, e.g., both eukaryotic and bacterial ribosomes.

1. Line 49: After the discussion of ribotoxins targeting SRL and before small molecule toxins inhibiting protein translation, the authors should add brief discussion of ribotoxins that target the decoding center.

2. Discussion section: It might be helpful to have a brief discussion why Ribocin does not cleave bacterial ribosome at H69. Sarcin is also eukaryotic ribosome specific. But Sarcin can also cleave bacterial ribosome, although the activity is significantly lower. Is there any data showing that Ribocin does not cleave bacterial ribosome at all?

---

## [Editor Report · Decision Letter 2]

10 Apr 2026

Dear Dr Vasquez-Rifo,

Thank you for your patience while we considered your revised manuscript "P. aeruginosa Ribocin cleaves ribosomal Helix 69 to inhibit host translation" for publication as a Update Article at PLOS Biology. This revised version of your manuscript has been evaluated by the PLOS Biology editors, and the Academic Editor.

Based on our Academic Editor's assessment of your revision, we are likely to accept this manuscript for publication, provided you satisfactorily address the remaining editorial points. Please also make sure to address the following data and other policy-related requests.

1) We routinely suggest changes to titles to ensure maximum accessibility for a broad, non-specialist readership, and to ensure they reflect the contents of the paper. In this case, we would suggest a minor edit to the title, as follows. Please ensure you change both the manuscript file and the online submission system, as they need to match for final acceptance:

“The Pseudomonas aeruginosa ribonuclease Ribocin cleaves eukaryotic ribosomes at helix 69 to inhibit host translation”

2) Thank you for providing already the raw data for some of the Figures. However, we are missing the raw data for Figures 1B, 4A, 5A; S4ABDEF, S6ABC, S7

3) Please cite the location of the data clearly in all relevant main and supplementary Figure legends, e.g. “The data underlying this Figure can be found in S1 Data” or “The data underlying this Figure can be found in https://doi.org/10.5281/zenodo.XXXXX”

4) Please ensure that you are using best practice for statistical reporting and data presentation. These are our guidelines https://journals.plos.org/plosbiology/s/best-practices-in-research-reporting#loc-statistical-reporting and a useful resource on data presentation https://journals.plos.org/plosbiology/article?id=10.1371/journal.pbio.1002128

-- If you are reporting experiments where n ≤ 5, please plot each individual data point.

5) Thank you for providing the original, uncropped and minimally adjusted images supporting some of the blots and gel results reported. I believe we are missing the uncropped images in Figures S3ABC

We will require these files before a manuscript can be accepted so please prepare and upload them now. Please carefully read our guidelines for how to prepare and upload this data: https://journals.plos.org/plosbiology/s/figures#loc-blot-and-gel-reporting-requirements

6) Please provide the tree files for the phylogenetic trees in Figures S1E. Please make sure all relevant figures have scale bars.

7) For figures containing any spectrometry data (Figures 2A, 3BD, S8A), please deposit the data on publicly available databases (see suggested db here; https://journals.plos.org/plosbiology/s/recommended-repositories) and provide the accession number/URL of the deposition in the Data Availability Statement in the online submission form.

8) Supplementary files (e.g., excel). Please ensure that all data files are uploaded as 'Supporting Information' and are invariably referred to (in the manuscript, figure legends, and the Description field when uploading your files) using the following format verbatim: S1 Data, S2 Data, etc. Multiple panels of a single or even several figures can be included as multiple sheets in one excel file that is saved using exactly the following convention: S1_Data.xlsx (using an underscore).

9) Please add a scale bar in the following microscopy pictures in Figures: 4F, 5BE, S4C, S5

10) We would like to enocurage you to deposit the plasmids constructed in this study and related research materials into the Addgene Repository

11) Please ensure that your Data Statement in the submission system accurately describes where your data can be found and is in final format, as it will be published as written there

12) Per journal policy, if you have generated any custom code during the course of this investigation, please make it available without restrictions. Please ensure that the code is sufficiently well documented and reusable, and that your Data Statement in the Editorial Manager submission system accurately describes where your code can be found. More information on our Code Policy, what and how to share can be found here: https://journals.plos.org/plosbiology/s/code-availability

We expect to receive your revised manuscript within two weeks.

*Published Peer Review History*

*Press*

Sincerely,

Melissa

Melissa Vazquez Hernandez, Ph.D.

Associate Editor

PLOS Biology

---

## [Editor Report · Decision Letter 3]

22 Apr 2026

Dear Dr Vasquez-Rifo,

Thank you for the submission of your revised Update Article "The Pseudomonas aeruginosa ribonuclease Ribocin cleaves eukaryotic ribosomes at helix 69 to inhibit host translation" for publication in PLOS Biology. On behalf of my colleagues and the Academic Editor, Matthew K. Waldor, I am pleased to say that we can in principle accept your manuscript for publication, provided you address any remaining formatting and reporting issues. These will be detailed in an email you should receive within 2-3 business days from our colleagues in the journal operations team; no action is required from you until then. Please note that we will not be able to formally accept your manuscript and schedule it for publication until you have completed any requested changes.

PRESS

Sincerely,

Melissa

Melissa Vazquez Hernandez, Ph.D., Ph.D.

Associate Editor

PLOS Biology
